



**Marked observed interannual differences in the vegetation response to the trend**
**towards a warmer and wetter climate in northwest China**
Shijun Zheng[1,2], Dailiang Peng[1,2,*], Bing Zhang[1,2,3,*], Yuhao Pan[1,2,3], Le Yu[4], Yan Wang[5],
Xuxiang Feng[6], Changyong Dou[1,2]
[1]Key Laboratory of Digital Earth Science, Aerospace Information Research Institute, Chinese
Academy of Sciences, Beijing 100094, China;
[2]International Research Center of Big Data for Sustainable Development Goals, Beijing 100094,
China;
[3]University of Chinese Academy of Sciences, Beijing 100049, China;
[4]Ministry of Education Key Laboratory for Earth System Modeling, Department of Earth System
Science, Tsinghua University, Beijing 100084, China;
[5]Department of Land Surveying and Geo-Informatics, The Hong Kong Polytechnic University,
Hong Kong 999077, China
[6]China Remote Sensing Satellite Ground Station (RSGS), Aerospace Information Research Institute,
Chinese Academy of Sciences, Beijing 100094, China
*Correspondence: Dailiang Peng (pengdl@aircas.ac.cn); Bing Zhang (zb@radi.ac.cn)





**Abstract**
Located in the interior of Eurasia, the Northwest China undergoes severe drought
as oceanic moisture is hard to travel a long distance and cross lots of mountain barriers.
These special geo-climatic conditions result in Northwest China being highly sensitive
to climate change. In this study, the response of the normalized difference vegetation
index (NDVI) to the trend towards a warmer and wetter climate in Northwest China
from 1982 to 2019 were investigated. The results show that there were significant
differences between the periods 1982–2000 and 2000–2019, with overall precipitation
decreasing before 2000 but increasing afterwards. After 2000, the temperature
increasing rate slowed down, whereas the NDVI increased at an obviously faster rate.
Compared with the period 1982–2000, the NDVI during the period 2000–2019 was
more affected by precipitation than by the temperature. The results of a normalized
linear regression also show that, for most vegetation types, the temperature played a
more dominant role during the period 1982–2000, whereas precipitation had a more
significant effect on the NDVI during the period 2000–2019. Throughout the study
period, the temperature had a greater impact on forest NDVI and the precipitation had
a greater impact on the NDVI in areas of bare land. In addition, the results show that
the strength of the relationship between the NDVI and climate in Northwest China
changed over time, with the relationship between NDVI and precipitation tending to
become stronger and the relationship between NDVI and temperature tending to
become weaker. The results will provide a new understanding of the relationship
between vegetation and climate in Northwest China and help to better cope with the
risks brought by climate change.
**Key words: Northwest China; vegetation; warmer and wetter climate; NDVI;**
**1. Introduction**
In arid areas, vegetation is affected by the limited precipitation and strong
evapotranspiration and is highly sensitive to climate change (Fensholt et al., 2009;



McGwire et al., 2000). There have been a large number of studies related to climate
change in Northwest China, and these show a clear trend towards warmer and wetter
conditions in the region in recent decades (Liu et al., 2013; Shi et al., 2002; Shi et al.,
2007; Wang et al., 2020; Wang et al., 2007; Zhang et al., 2021; Zheng et al., 2021).
The relationship between vegetation and climate is very complex. On the one hand,
vegetation is affected by the climate, and the spatial heterogeneity of vegetation growth
is closely related to the climate conditions. For example, the spatial patterns in
vegetation productivity in grassland areas where water is scarce are consistent with the
spatial patterns in annual precipitation (Piao et al., 2006). However, vegetation can
continuously adapt to changes in climate conditions: for example, it was found that the
precipitation threshold for vegetation growth in Australia decreased from 1982 to 2010
(Ukkola et al., 2016). In addition, some studies have found that the relationship between
temperature and vegetation productivity may change over time given other
environmental limitations (Angert et al., 2005; Beck and Goetz, 2012). Thus, the impact
of climate on vegetation is a dynamic process and may change over time.
There have been many studies on the relationship between vegetation and climate
in Northwest China, but most of which have been carried out over different time scales
and usually focused on a fixed short period of time, which resulted in different results.
(Cao et al., 2011; Guo et al., 2008; Xiu-hua et al., 2009; Zhang et al., 2016; Zhao et al.,
2011). As climate varies with time, its impact on vegetation growth varies in different
time periods. In addition, most these studies analyzed all vegetation as a whole, while
few have distinguished between different vegetation cover types. This study aimed to
address these problems in earlier studies. The discrepancy in the NDVI response to
climate change for different periods within the overall study period were analyzed for
different vegetation types.



**2. Methodology**

**2.1 Study area**

This study focuses on northwest China (Figure 1). Located in the interior of Eurasia, the altitude of this region ranges from –152 m to 8058 m, with most areas lying at an altitude of over 1000 m. The complex terrain means that the oceanic moisture is hard to travel a long distance and cross the mountain barriers, resulting in a dry climate.

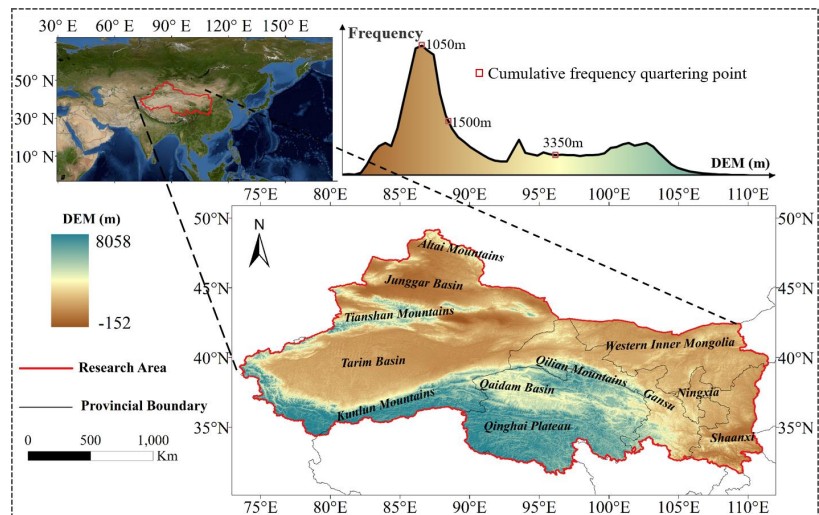

**Figure 1. Study area and elevation**

**2.2 Data and methods**

**2.2.1 NDVI time series**

In this study, combined GIMMS NDVI and MODIS NDVI data acquired during the period 1982–2019 were used. The GIMMS NDVI data were acquired at 15-day intervals during the period 1982–2015 and had a spatial resolution of 8 × 8 km; these data were provided by the National Natural Science Foundation of China's Environmental and Ecological Science Data Center for West China





(http://data.tpdc.ac.cn/zh-hans/data/1cad1a63-ca8d-431a-b2b2-
45d9916d860d/?q=GIMMS). MODIS NDVI data for 2000–2019 that had a spatial
resolution of 250 × 250 m and that were also acquired at 15-day intervals were
provided by NASA (https://reverb.echo.nasa.gov). Both datasets were composited into
monthly values using the maximum value composite (MVC) technique. The NDVI
during the growing season was calculated by averaging the values for the period April
to October. A complete NDVI time series for 1982 to 2019 that had a spatial resolution
of 8 × 8 km was constructed using pixel-wise linear regression. The new 2016-2019
was called the expanded NDVI. To check the accuracy of the constructed long-term
time series, statistical analysis was performed on the GIMMS NDVI and the extended
NDVI. The results are shown in Fig 2.

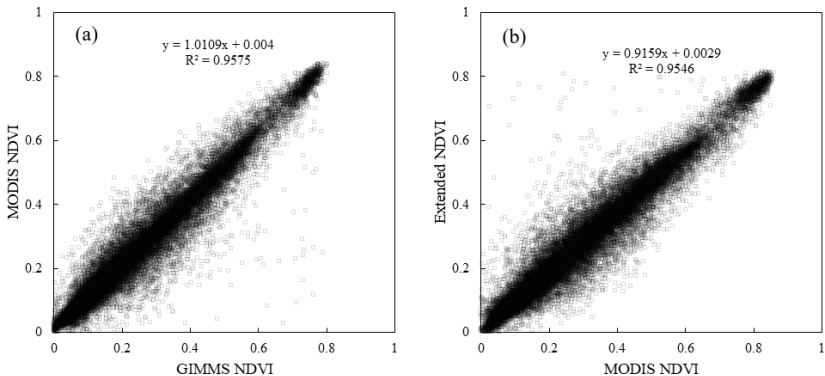


**Figure 2. Results of the consistency checks for the different datasets: (a) MODIS NDVI pixel**
**values plotted against GIMMS NDVI pixel values; (b) MODIS NDVI pixel values plotted**
**against extended NDVI pixel values**
**2.2.2 Climate variable time series**
The time series of gridded climate variable data (precipitation and temperature)
used in this study were obtained by applying multiple regression and residual
interpolation at a spatial resolution of 1 × 1 km to meteorological station observations
(Zheng et al., 2021). The data has been verified to have a high accuracy by comparing
with the meteorological station data (Zheng et al., 2021).





**2.2.3 Other data**
The    land    cover    data    originated    from    Tsinghua    University.    See
(https://www.resdc.cn/Datalist1.aspx?FieldTyepID=1,3)   for   more   information. The
afforestation data for Northwest China were obtained from the National Bureau of
Statistics (http://www.stats.gov.cn) and the population and GDP data were obtained
from the Data Center for Resources and Environmental Sciences, Chinese Academy of
Sciences (http://www.resdc.cn).
**2.2.5 Partial correlation analysis**
Partial correlation coefficients can be defined by equations (6) and (7) (Kenett et
al., 2015; Liu et al., 2015; Song and Ma, 2011):
$$r_{xy} = \frac{\sum_{i=1}^{n}(x_i-\bar{x})(y_i-\bar{y})}{\sqrt{\sum_{i=1}^{n}(x_i-\bar{x})^2 \sum_{i=1}^{n}(y_i-\bar{y})^2}} \tag{6}$$

and
$$r_{xy,z} = \frac{r_{xy}-r_{xz}r_{yz}}{\sqrt{(1-r_{xz}^2)(1-r_{yz}^2)}}, \tag{7}$$

where $r_{xy}$, $r_{xz}$, $r_{yz}$ are the correlation coefficients between variables $x$ and $y$, $x$
and $z$, and $y$ and $z$, respectively; $\bar{x}$ and $\bar{y}$ are the average of $x$ and $y$, respectively; and
$r_{xy,z}$ is the partial correlation coefficient between $x$ and $y$, excluding the influence of $z$
on $x$ and $y$.
In this study, the statistical significance of the partial correlations was calculated
using the t-test (equation (8)) (Song and Ma, 2011) with the significance level set to

125     0.05:

$$t_{xy,z} = \frac{r_{xy,z}}{\sqrt{1-r_{xy,z}^2}}\sqrt{n-m-1}. \tag{8}$$

Here, $m$ is the number of independent variables (m = 2 in the study) and $n$ is
the number of samples.



**2.2.6 Normalized linear regression**
For a further comparison of the impact of the temperature and precipitation on the
NDVI, a multivariate linear regression was conducted. The NDVI was then given as
$$NDVI = b_0 + b_1 \times Temperature + b_2 \times Precipitation + \varepsilon. \qquad (9)$$
Here, $NDVI$ represents the value of the NDVI during the growing season, $b_0$ is
the intercept of the regression model, and $b_1$ and $b_2$ are the regression coefficients for
the temperature and precipitation, respectively. ε is the regression residual.
In addition, due to the dimensional difference between the temperature and
precipitation, a normalization was carried out so that the importance of the influence of
these two climate factors on the NDVI could be compared. After the normalization, the
larger the absolute value of the regression slope, the more significant the impact of the
corresponding independent variable on the NDVI. The normalization can be
represented by
$$var_{nor} = \frac{var - var_{min}}{var_{max} - var_{min}}, \qquad (10)$$
where $var$ is the variable to be normalized – in this study, the NDVI, temperature,
or precipitation; $var_{min}$ and $var_{max}$ are the minimum and maximum value,
respectively, of the variable in the time series; and $var_{nor}$ is the normalized value of
$var$.
**3. Results and Discussion**
**3.1 Impact of climate variables on the NDVI**
Significant differences were also found in the spatial characteristics of the trends
in the precipitation, temperature, and NDVI before and after 2000 (Figure 3). Before
2000, in most parts of the study region, including southern Qinghai, southern Gansu,
and the whole of Shaanxi, the precipitation shows a downward trend, whereas, after
2000, it increases in most of these areas. Although the temperature increases during

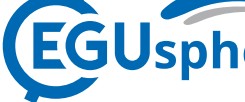

both periods, the rate of increase before 2000 is significantly greater than that after 2000,
with more regions passing the significant test. Compared with before 2000, in the areas
where there is an increasing NDVI trend, the trend after 2000 is more significant. The
same is true where the NDVI trend is downwards, particularly in the Junggar basin,
Tianshan and southern Qinghai.

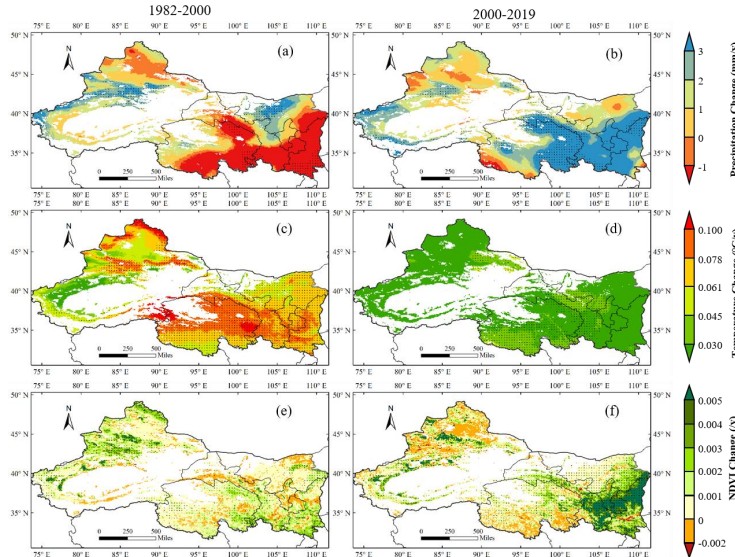


**Figure 3. Spatial characteristics of the trends in (a) precipitation, (c) temperature, and (e)**
**NDVI for 1982–2000; spatial characteristics of the trends in (b) precipitation, (d)**
**temperature, and (f) NDVI for 2000–2019 for northwest China during the growing season.**
**The areas marked with dots pass the significance test (S < 0.05)**

165          To study effects of climate change on the NDVI, we calculated the partial

correlation coefficients between the NDVI and precipitation and NDVI and temperature
for each pixel in the study area for the periods 1982–2000 and 2000–2019. The results
are shown in Figure 4. For 1982–2000, the areas with a positive correlation between
the NDVI and precipitation are mainly located in the Tianshan Mountains, the western
part of Xinjiang, the southern parts of Gansu and Ningxia, and the western part of Inner
Mongolia; the areas where the correlation is negative are mainly located in Qinghai and
the southern part of Shaanxi. The main areas where there is a positive correlation



between the NDVI and temperature are found in the Altai Mountains, the western part
of Xinjiang, the northern and southern parts of Qinghai and northern Shaanxi; the main
areas where there is a negative correlation are located in the Qilian Mountains, western
Inner Mongolia and southern Ningxia. For the period 2000–2019, there is a significant
increase in the extent of the areas where there is a significant positive correlation
between the NDVI and precipitation: these areas are mainly in the east of the study
region; in contrast, there is a decrease in the extent of the areas where there is a positive
correlation between the NDVI and temperature. In addition, the correlation between the
NDVI and precipitation and the NDVI and temperature exhibit opposite characteristics
in many regions: the NDVI is negatively correlated with precipitation but positively
correlated with temperature in southern Qinghai and southern Shaanxi, which are areas
that either are at a high altitude or have a high annual precipitation. However, in the
Junggar Basin and the areas surrounding the Tarim Basin, which have a dry climate,
the NDVI is positively correlated with precipitation but negatively correlated with
temperature.

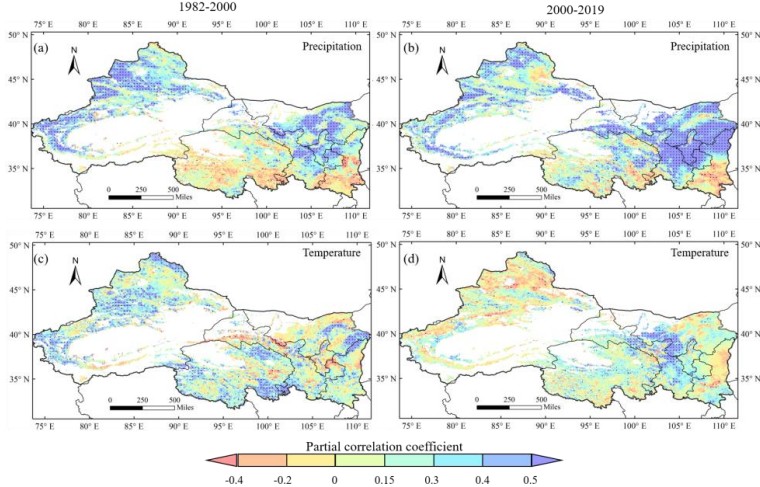


**Figure 4. Spatial characteristics of the partial correlation coefficients between the NDVI and**
**(a) precipitation and (c) temperature for 1982–2000 and between the NDVI and (b)**
**precipitation and (d) temperature for 2000–2019 in northwest China. The area marked with**
**dots pass the significance test (S < 0.05)**




Frequency statistics were calculated for the partial correlation coefficients between
the gridded NDVI values for different vegetation types and temperature and between
these values and precipitation. The results are shown in Figure 5. It can be seen that the
response of the NDVI to temperature and precipitation in northwest China varies
greatly between the two parts of the study period as well as by vegetation type. For
cropland and grassland, during the period 1982–2000, the temperature and precipitation
had a similar effect on the NDVI, whereas during the period 2000–2019, the NDVI was
affected more by the precipitation than the temperature. For the forest land-cover type,
the influence of the temperature on the NDVI was more significant than that of the
precipitation during both periods. For shrubland, the NDVI was more affected by
temperature during the period 1982–2000 but more affected by the precipitation during
the period 2000–2019. For the sparse vegetation in areas of bare land, the impact of
precipitation on the NDVI were greater than that of temperature during both periods.
On the whole, compared with 1982–2000, for all vegetation types, the NDVI was more
affected by precipitation during the period 2000–2019 and less affected by the
temperature.

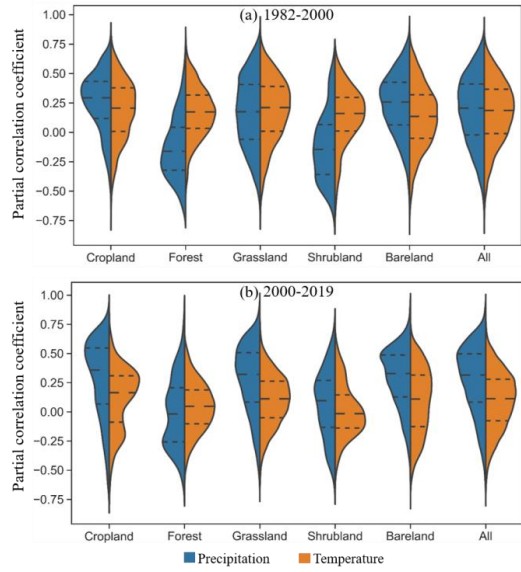






**Figure 5. Statistics relating to the frequency distribution of the partial correlation**
**coefficients between NDVI and precipitation and NDVI and temperature for different**
**vegetation cover types during the periods (a) 1982–2000 and (b) 2000–2019 in northwest**
**China**

In order to more comprehensively analyze the impact of the climate on the NDVI
in northwest China, a normalized linear regression analysis was carried out on the
relationships between the annual regional average NDVI and the temperature and the
annual average NDVI and the precipitation. The results are shown in Figures 6 and 7.
From 1982 to 2000, with the exception of the bare land class,   the temperature
regression coefficient is greater than that of the precipitation coefficient, which
indicates that the temperature had a more dominant influence on vegetation in
northwest China during this period, whereas during the period 2000–2019, the
influence of the precipitation was more dominant. During this latter period, with the
exception of forest areas, the precipitation regression coefficient is greater than the
temperature regression coefficient. However, during both periods, the temperature had
a greater impact on the forest areas than the precipitation did, whereas in areas of bare
land less it had a smaller impact. These results are consistent with the results of the
partial correlation analysis described above.



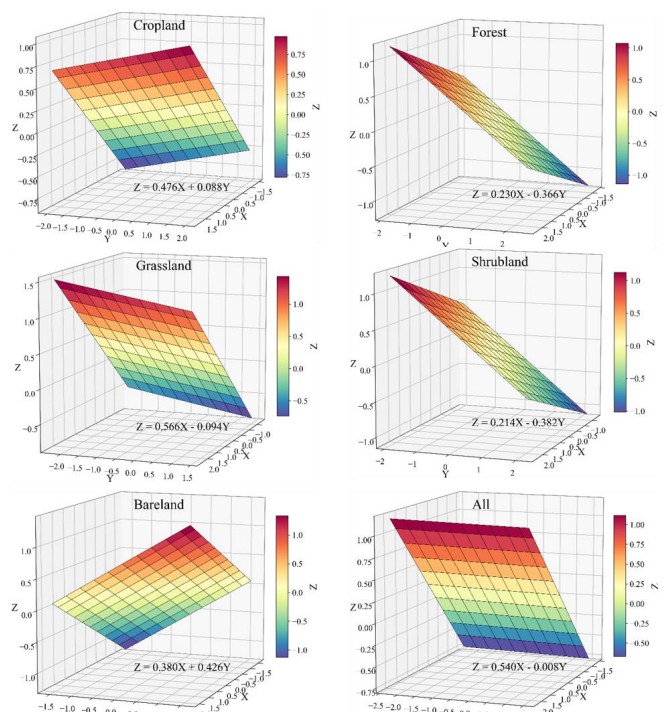

**Figure 6. Normalized linear regression results for the relationships between the NDVI and temperature and NDVI and precipitation for different vegetation types in northwest China from 1982 to 2000. X represents the normalized temperature, Y the normalized precipitation, and Z the normalized NDVI.**





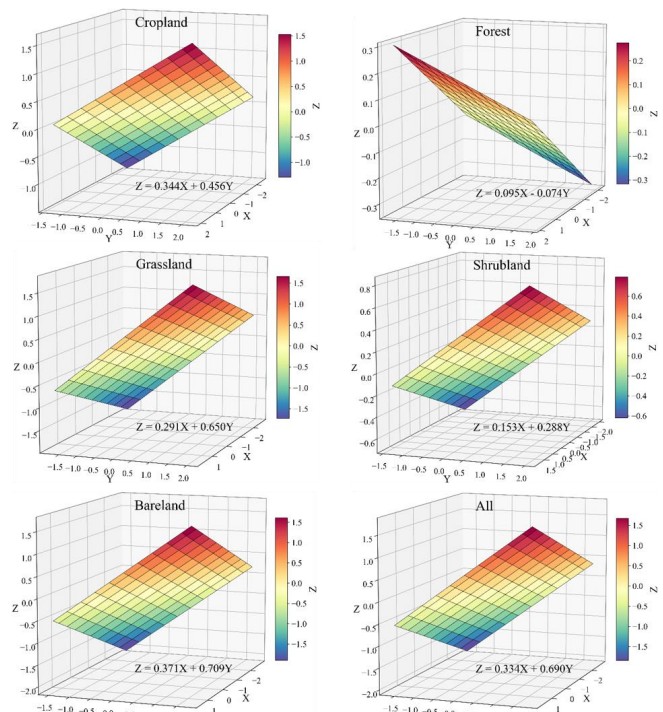

**Figure 7. Normalized linear regression results for the relationships between the NDVI and temperature and NDVI and precipitation for different vegetation types in northwest China from 2000 to 2019. X represents the normalized temperature, Y the normalized precipitation, and Z the normalized NDVI.**

## 3.2 Characteristics of the response of vegetation to climate change over time

According to the above analysis, it is clear that the impact of the temperature and precipitation on vegetation in northwest China differed between the two periods 1982–2000 and 2000–2019. To explore whether there were interannual variations in the relationships between the temperature and vegetation greenness and precipitation and vegetation greenness, the partial correlation coefficients for the relationships between the regional average NDVI and temperature and the regional average NDVI and precipitation were calculated for the different vegetation types for the period 1982–2019 using a nine-year sliding window. The results of this are presented in Figure 8. For forest, grassland, and shrubland, the correlation between the NDVI and temperature



shows an obvious downward trend, indicating that the positive impact of the
temperature on these types of vegetation was weakening over time. In contrast, for the
grassland and shrubland classes, the correlation between the NDVI and precipitation
increases over the study period, indicating that the precipitation was having an
increasing influence on the vegetation. For areas of bare land, there is an obvious
turning point in the correlation between the NDVI and temperature and between the
NDVI and precipitation: before 1999, there is a significant decrease in the correlation
between the NDVI and precipitation, whereas there is a significant increase in the
correlation between the NDVI and temperature; after 1999, both these trends change
direction. Overall, the strength of the relationship between the NDVI and precipitation
in northwest China became stronger over the years 1982–2019, whereas the relationship
between the NDVI and temperature weakened.

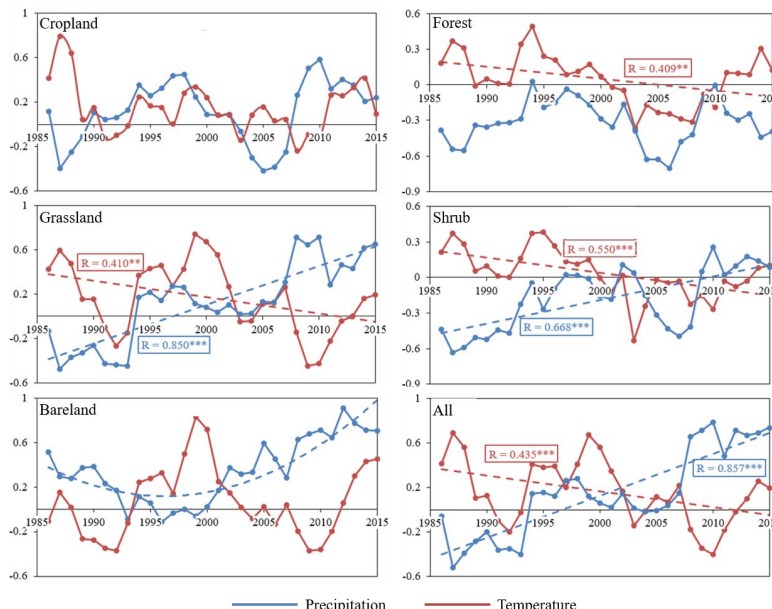


**Figure 8. Partial correlation coefficients for the relationships between the regional nine-year**
**average NDVI and temperature and the nine-year average NDVI and precipitation for**
**different vegetation types in northwest China for the period 1982–2019. R represents the**
**partial correlation coefficient: values are given for relationships exhibiting significant**
**changes only.**

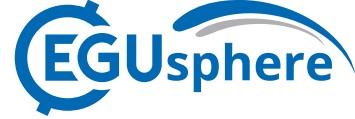

By calculating the partial correlation coefficients between the NDVI and
temperature and NDVI and precipitation for each pixel using a nine-year sliding
window, the spatial characteristics of the trends in these partial correlation coefficients
were obtained. The results are shown in Figure 9. It can be seen that in most parts of
northwest China, there is a downward trend in the correlation between the NDVI and
temperature, with this trend being significant for 30% of pixels; there is an upward trend
in the correlation between the NDVI and precipitation, with this trend being significant
for 32% of pixels. It is generally agreed that vegetation will continue to adapt to climate
change in order to grow better (Ukkola et al., 2016). As the climate of northwest China
has become warmer and wetter in recent years, the response of the vegetation in this
region to temperature and precipitation has changed, which means that the potential
risks brought by future climate change cannot be ignored.

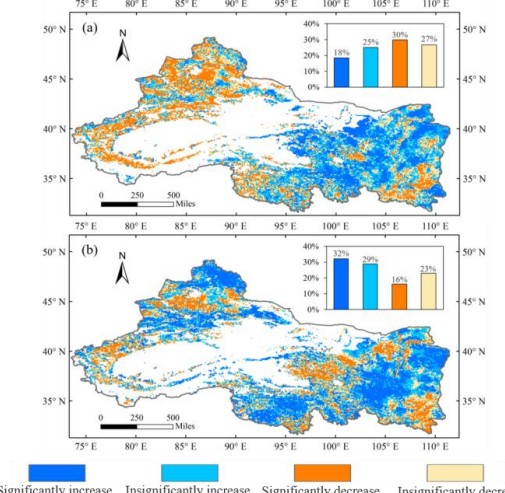


**Figure 9. Spatial distribution of significant changes in partial correlation coefficients**
**between the NDVI and (a) temperature and (b) precipitation in northwest China, calculated**
**using a nine-year sliding window for the period 1982–2019**

**3.3. Disturbances due to human activity and uncertainties affecting the analysis**

286    In addition to the influence of climate change, human activities (urbanization,
287    industrialization, afforestation, etc.) also have an important impact on vegetation (Guan



et al., 2018; Lin et al., 2020; Wang et al., 2012; Xu et al., 2010). The Chinese government has implemented a series of afforestation programs and measures to reduce desertification in northwest China, including the Three-North Shelter Forest Program (1991), the Natural Forest Protection Program (1998), the Cropland to Forest and Grassland Conversion Program (1999), the Law on Desert Prevention and Transformation (2001), the Beijing–Tianjin–Hebei Sandstorm Source Control Project (2002), as well as a project forbidding grazing established in 2003. As a result, the area affected by both desertification and sandy desertification in China has continued to decline since 1999 (Figure 10). An analysis of artificial forests in five provinces in northwest China (Figure 11) showed that the total area covered by these forests is around 260000 km$^2$; 60000 km$^2$ of this is in Gansu and Shaanxi provinces. The correlation between the NDVI and the total area of artificial forest was found to be weakest in Qinghai, which has the smallest total area of artificial forest among the provinces analyzed. In Gansu and Shaanxi, the NDVI is greatly influenced by the artificial forests, and the correlation there is greater than 0.80. It appears that the amount of vegetation greenness in Ningxia is also strongly influenced by these artificial forests, and the correlation between the changes in the NDVI and the total area of artificial forest is particularly strong here both before and after 2000.

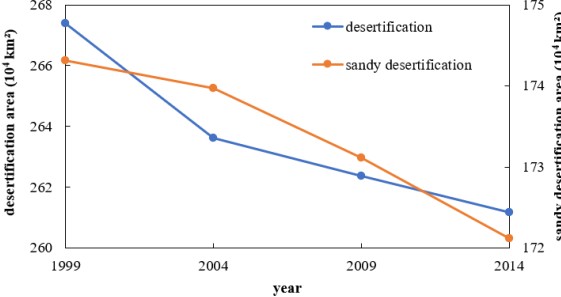

**Figure 10. Area affected by desertification (left-hand y-axis) and sandy desertification (right-and y-axis) in China over the period 1999–2014. These data were derived from four national desertification and sandy desertification monitoring bulletins.**





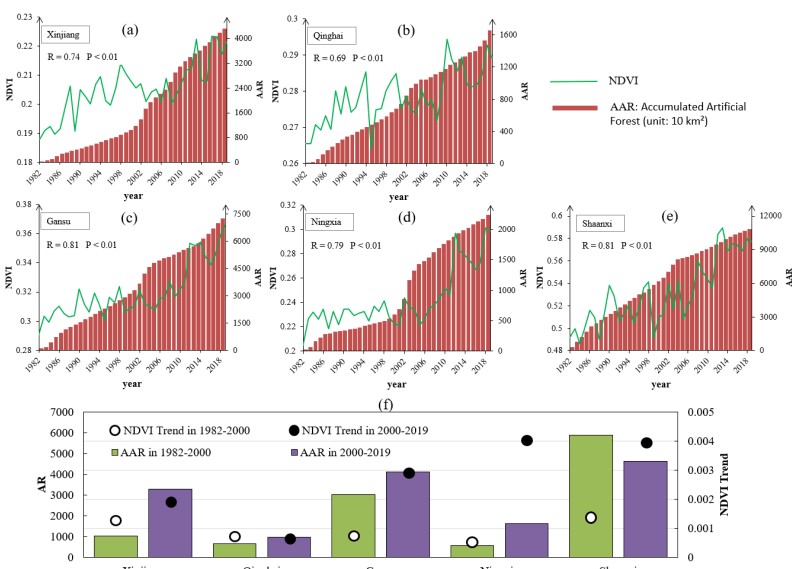

**Figure 11. Regional average NDVI and total area of artificial forest in five different provinces in northwest China for the period 1982–2019 (R is the correlation coefficient; P is the significance of the linear relationship between the NDVI and AAR (Accumulated Artificial Forest)); (a)–(e) show data for Xinjiang, Qinghai, Gansu, Ningxia, and Shaanxi, respectively. (f) shows the AAR and the trend in the average NDVI for the same five provinces in northwest China during the periods 1982–2000 and 2000–2019.**

From 1995 to 2015, although the total population of northwest China increased, there was a decrease in the rural population as a result of migration to urban centres (Figure 12a); this has tended to lessen the pressure on natural vegetated land and may have contributed to an increase in the NDVI over a large part of the region (Yuan et al., 2019). In addition, a larger population means that agricultural oases are expanding – this is considered to be an important factor promoting the greening of vegetation in northwest China (Jiapaer et al., 2015; Xie et al., 2018). However, the GDP of most areas in the study region increased over the study period (Figure 12b), possibly leading to overgrazing and land degradation (Yuan et al., 2019). Moreover, rapid urbanization can also lead to a reduction in ecosystem services (Guan et al., 2018), which adversely affects vegetation.

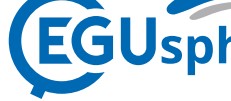

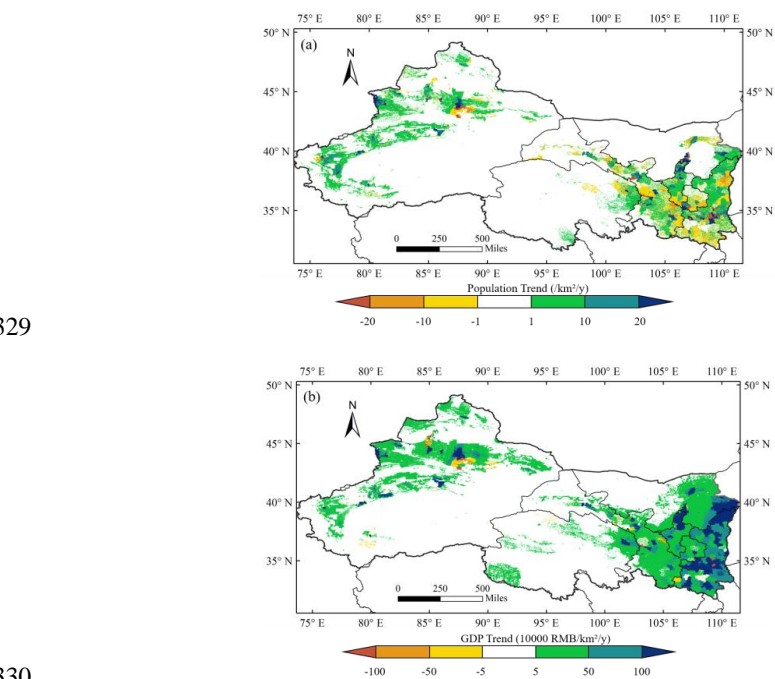

**Figure 12 Spatial characteristics of the trends in (a) population and (b) GDP from 1995 to 2019**

In this study, both a partial correlation analysis and normalized linear regression were carried out based on the linear correlation between the NDVI and temperature and the NDVI and precipitation. However, the response of vegetation to climate change may be non-linear. Also, the impact of climate on vegetation may not be immediate and could have a time lag ranging from one to several months. These questions have not yet been investigated in great depth and deserve further exploration.

**4. Conclusions**

In this study, the spatiotemporal trends in the NDVI, precipitation, and temperature were analyzed for different vegetation types in northwest China for the period 1982 to 2019. A linear regression analysis indicated that the details of these trends for 1982–2000 were significantly different from those for 2000–2019, with precipitation

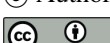



decreasing before 2000 but increasing afterwards, the temperature growth slowing after
2000, and the NDVI increasing significantly faster. These overall trends were also
reflected in the trends for most of the individual vegetation types.
Before 2000, the impact of the precipitation on cropland and grassland was similar
to that of the temperature; the temperature had a greater impact on forest and shrubland
than the precipitation, while the precipitation had a greater impact on areas of bare land.
After 2000, except for the forest class, the positive correlation between the precipitation
and vegetation greenness was stronger than that between the temperature and vegetation
greenness. In addition, compared with 1982–2000, during the period 2000–2019, the
NDVI of all vegetation types was clearly more affected by the precipitation and less
affected by the temperature. To gain a more comprehensive understanding of the impact
of climate on the NDVI in northwest China, we carried out a normalized linear
regression analysis and found that the temperature played a more dominant role during
the period 1982–2000 whereas the precipitation played a more dominant role during
the period 2000–2019, which was consistent with the partial correlation results.
To explore the characteristics of the vegetation response to climate change over
time, we conducted a partial correlation analysis between the NDVI and the climate
variables using a nine-year sliding window and found that the relationship between the
NDVI and precipitation became stronger, whereas the relationship between the NDVI
and temperature became weaker during the period 1982 to 2019 in northwest China.
In addition to the effects of climate change, we also studied the disturbance to
vegetation caused by human activities. We found a strong correlation between the total
area of artificial forest and the NDVI, indicating the great positive impact of
afforestation programs on vegetation in northwest China.
**Data and code availability**
GIMMS NDVI and MODIS NDVI data are available in http://data.tpdc.ac.cn/zh-
hans/data/1cad1a63-ca8d-431a-b2b2-45d9916d860d/?q=GIMMS                    and



https://reverb.echo.nasa.gov/, respectively. The land cover data originated from
Tsinghua University. See (https://www.resdc.cn/Datalist1.aspx?FieldTyepID=1,3) for
more information. The afforestation data for Northwest China were obtained from the
National Bureau of Statistics (http://www.stats.gov.cn) and the population and GDP
data were obtained from the Data Center for Resources and Environmental Sciences,
Chinese Academy of Sciences (http://www.resdc.cn). All the analyses are made using
Python, and the code are available from the corresponding author on reasonable request.

**Author contributions**

Shijun Zheng, Dailiang Peng and Bing Zhang designed the research ideas. Shijun
Zheng and Yan Wang completed the code for analysis. Yuhao Pan prepared the NDVI
and climate data. Le Yu helped with the preparation of the land cover datasets. Xuxiang
Feng prepared the afforestation data. Changyong Dou prepared the population and GDP
data. All authors commented on the paper and provided feedback throughout the data
analysis.

**Competing interests**

The contact author has declared that none of the authors has any competing
interests.

**Acknowledgment**

This study was supported by the National Natural Science Foundation of China
(grant number 42071329).

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
