# Peer review of "Marked observed interannual differences in the vegetation response to the trend"

_EGUsphere, 2022_

## Author Comment (AC1)

**Responses and Changes to Reviewers' Comments**

**Dear Reviewers,**

We really appreciate your helpful suggestions and comments. We have carefully revised the manuscript and addressed all comments. In terms of content, we mainly increased the experimentation of CRU and ERA5 meteorological data to enhance the reliability of the article results. We also conducted a mechanism analysis to investigate how drought regulates the relationship between vegetation and precipitation (temperature). As for the methodology, we detrended all variables before studying the vegetation-climate relationship to prevent statistical analysis independence. Instead of comparing the NDVI and climate relationship between pre- and post-2000, we used multiple sliding windows to emphasize the inter-annual variability of this relationship. In writing, we improved the language use throughout the entire article, made significant changes to the introduction to highlight the innovation of this study, and added descriptions of the interpretation of the results, as well as discussions of the results.

**Referee: 1**

1. **The description of Introduction is inadequate. For example, no sufficient evidence is provided to support why Northwest China is selected . In addition, I do not fully agree with the author's statement that previous studies pay little attention to the long term changes of vegetation growth to climate change in Northwest China, as this region is usually included in a larger spatial extent, such as northern China, Central Eurasia or even the drylands of the Northern Hemisphere. Meanwhile, the diverse response of vegetation growth to climate variables across land surfaces has always been a hot topic and many interesting findings are found. Above all, the summary of previous studies is insufficient and arbitrary. As a result, the author is unable to give a clear scientific hypothesis.**

   **Response:**

   Thank you very much for your important suggestions. We have made significant changes to the introduction.

   We added an explanation in the introduction about why Northwest China was selected as the study region: "Northwest China is characterized by vast areas with different land cover types, including grasslands, forests, and barren lands with sparse vegetation. Since the early 1980s, several studies have indicated warmer and more humid conditions in this area (Liu et al., 2013; Shi et al., 2002; Shi et al., 2007; Wang et al., 2020; Wang et al., 2007; Zhang et al., 2021; Zheng et al., 2021). Recent decades have also seen significant changes in the growth of vegetation in this region (Chen et al., 2019; Niu et al., 2019). As a result, Northwest China presents an ideal opportunity for examining the changes in relationship between climate and vegetation across a variety of vegetation types".

   We have added a detailed description of previous studies: "The changing correlation between climate and vegetation has recently gained some attention. For example, Wang and Yan (2021) found that the correlation between vegetation and

temperature has weakened throughout China over the past 34 years. The precipitation threshold required for vegetation growth in Australia had been found to decrease from 1982 to 2010 (Ukkola et al., 2016). Keenan and Riley (2018) measured how vegetation cover responded to temperature changes and found that the limitations imposed by temperature had decreased over time. Zhao and Yu (2021) found an increased association between climate change and vegetation index variation in Northwest China over the past 34 years. However, most of these studies have solely identified the occurrence of the changes in the relationship between precipitation (or temperature) and vegetation. Uncertainties remain regarding the drivers and how they regulate the changes in the relationship"

The diverse response of vegetation growth to climate variables across land surfaces has indeed always been a hot topic. However, variations may also exist in the patterns of changes in the relationship between climate and vegetation across distinct types of land cover, and the mechanisms behind them are also different, which has been confirmed by our results, but this area of research has not received enough attention yet. Therefore, we provided a detailed description in the introduction: " Vegetation greenness patterns display high spatial heterogeneity across different land surfaces (Gao et al., 2017; Wang et al., 2021), and it's response to climate also varies greatly among different terrestrial ecosystems (Yuan et al., 2019a). The influencing mechanisms of vegetation dynamics in diverse vegetation types have been well documented (Cai et al., 2021; Li et al., 2019; Luo and Chen, 2013; Tao et al., 2015; Upgupta et al., 2015; van Oijen et al., 2018; Wu et al., 2021). Based on previous research, it can be easily inferred that variations may also exist in the patterns of changes in the relationship between climate and vegetation across distinct types of land cover, and the mechanisms behind them are also different, but this area of research has not received enough attention yet"

2. **According to the results, the authors say the year of 2000 is an important turning point in time. However, there is no method description for defining the time turning point. It is unclear whether the turning point is robust and varies in space.**

**Response:**

Thank you very much for your suggestion. Instead of comparing the NDVI and climate relationship between pre- and post-2000, we now utilize multiple sliding windows to emphasize the inter-annual variability of this relationship

3. **The authors are suggested to add statistical analysis to compare the correlation coefficient of NDVI with temperature and precipitation, such as the results shown in Figure 5.**

**Response:**

Thank you very much for your suggestion. Based on three sets of meteorological data, we have compared the correlation of NDVI-temperature and the correlation of NDVI-precipitation, shown as below:

[Figure]

**Figure 3. Scatter plots between $R_{NDVI-GP}$ and precipitation from Interp (a), CRU (b) and ERA5 (c), and scatter plots between $R_{NDVI-GT}$ and temperature from Interp (d), CRU (e) and ERA5 (f).** Here, $R_{NDVI-GP}$ is the partial correlation coefficient between GS NDVI and GS precipitation, and $R_{NDVI-GT}$ is the partial correlation coefficient between GS NDVI and GS temperature (All variables are detrended). The yellow dashed line is the quadratic fit for all the scatters.

Then, we compared the trends of the correlation of NDVI-temperature and the correlation of NDVI-precipitation based on multiple sliding windows, The results with a 13-year sliding window are shown below, and the results of additional experiments conducted with sliding windows of 9, 11, 15, and 17 years are displayed in Supplementary Figs. 3~6.

[Figure]

**Figure 4. Changes in RNDVI-GP and RNDVI-GT at a 13-year sliding window**. Here $R_{NDVI-GP}$ (or $R_{NDVI-GT}$) is the partial correlation coefficient between GS NDVI and GS precipitation (or temperature), and is calculated using a 13-year sliding window during 1982-2015. All variables are detrended. The blue (or red) line represents the changes in $R_{NDVI-GP}$ (or $R_{NDVI-GT}$), and the shaded portion represents the confidence interval. Three types of climate data (Interp, CRU and ERA5) are used, corresponding to three columns. The first to fourth lines correspond to the following areas respectively: all vegetation areas except cropland, forest, grassland, and barren land (As the parts with NDVI<0.1 are removed, the barren areas here represent sparse vegetation). The symbol of *, ** and *** in the upper right of the value of r indicate the significant trend at P < 0.1, P < 0.05 and P < 0.01 respectively.

.

4. **It is unclear why a nine-year sliding window is used to show the time-varying sensitivity of NDVI to temperature and precipitation. How to consider the impact of solar radiation on vegetation growth? Is it a major driver?**

**Response:**

Thank you very much for your suggestion. We also realized that using only one sliding window to obtain results lacks scientific validity, so we used multiple sliding windows of 9, 11, 13, 15, and 17 years to enhance the robustness of our findings. The results indicated that the conclusions from different sliding windows were relatively consistent.

Due to the significant impact of radiation on vegetation growth, it was unscientific for us to ignore radiation when studying the correlation between NDVI

and precipitation (or temperature). Therefore, when calculating the partial correlation between NDVI and precipitation (or temperature), we also controlled for radiation to eliminate its interference, and we explained this in our methodology: "The relationship between precipitation (temperature) and NDVI is calculated as the partial correlation coefficient between GS precipitation (temperature) and GS NDVI, after statistically controlling for interannual variation in GS temperature (precipitation) and GS radiation". In addition, we also conducted a separate analysis of the relationship between radiation and NDVI, and the results are shown below, which is displayed in Supplementary Fig. 2. Meanwhile, we have added corresponding analyses in the article:" For some forested areas in Shaanxi, NDVI shows a weak negative correlation with precipitation (Figure 2a, c, e), which may be attributed to an increase in solar radiation as precipitation decreases in this region (Supplementary Fig. 1b, e). In order to explore how solar radiation affects vegetation productivity, we analyzed the partial correlations between interannual fluctuations in shortwave radiation and NDVI (Supplementary Fig. 2). The effect of radiation on plants also exhibits significant spatial variability. While enhanced radiation can boost photosynthesis efficiency and stimulate vegetation growth, it can also raise transpiration rates, resulting in soil moisture loss (Piao et al., 2014). NDVI and radiation exhibit the significant positive correlation in forests, while in high-altitude grasslands they mainly exhibit a negative correlation (Supplementary Fig. 2)"

[Figure]

**Supplementary Figure 2. Spatial distribution of partial correlation of NDVI and radiation.** All variables are detrended. To calculate the partial correlation versus GS radiation, GS temperature and precipitation are controlled for. The dots indicates the regions with significant relationship in $R_{NDVI-GR}$ (P < 0.05)

5. **Actually it is difficult to integrate data of different spatial scales. For example, the NDVI data is at the pixel scale; however, the social statistical data is at the county or even provincial scales. It is questionable whether the human activity rather than climate can play a dominant role in shaping regional NDVI. Besides, how to distinguish the effects of afforestation on NDVI, as the analysis of land-use and-cover changes are missing in this study. The relationship between NDVI and social-economic divers such as GDP and population is very complex that should not be the focus of this study. I think the authors should focus on the topic why the response of vegetation NDVI**

**changes over time. Ecosystem adaptation (e.g. changes in vegetation structure) or changes in environmental conditions, such as background soil moisture**

**Response:**

Thank you very much for your suggestions. Considering the complexity of human influence, we removed this part of the content in the article. In the discussion section, we analyzed how drought regulates the relationship between NDVI and precipitation (or temperature) based on two indicators, vapor pressure deficit (VPD) and soil water volume (SWV), and the results with a 13-year sliding window are shown below. We found that the fluctuations in $R_{NDVI-GP}$ and $R_{NDVI-GT}$ coincides closely with the variations in drought conditions. In the areas with the trend in VPD less than 0.02 hpa/yr, where grasslands are predominantly distributed, an increase in SWV tends to cause a decrease in $R_{NDVI-GP}$, but an increase in $R_{NDVI-GT}$. However, when the VPD trend exceeds 0.02 hPa/yr, a more negative trend in SWV tends to result in more negative trends in both $R_{NDVI-GP}$ and $R_{NDVI-GT}$.

[Figure]

**Figure 6. Average trend of $R_{NDVI-GP}$ and $R_{NDVI-GT}$ at a 13-year sliding window in a climate phrase space.** Here, $R_{NDVI-GP}$ is the partial correlation coefficient between GS NDVI and GS precipitation from Interp (**a**), CRU (**b**), and ERA5 (**c**); $R_{NDVI-GT}$ is the partial correlation coefficient between GS NDVI and GS temperature from Interp (**d**), CRU (**e**), and ERA5 (**f**). They are calculated using a 13-year sliding window during 1982-2015. All variables are detrended. The climate space is delineated by changes in GS soil water volume (SWV) and changes in vapor pressure deficit (VPD).

---

## Author Comment (AC2)

**Responses and Changes to Reviewers' Comments**

**Dear Reviewers,**

We really appreciate your helpful suggestions and comments. We have carefully revised the manuscript and addressed all comments. In terms of content, we mainly increased the experimentation of CRU and ERA5 meteorological data to enhance the reliability of the article results. We also conducted a mechanism analysis to investigate how drought regulates the relationship between vegetation and precipitation (temperature). As for the methodology, we detrended all variables before studying the vegetation-climate relationship to prevent statistical analysis independence. Instead of comparing the NDVI and climate relationship between pre- and post-2000, we used multiple sliding windows to emphasize the inter-annual variability of this relationship. In writing, we improved the language use throughout the entire article, made significant changes to the introduction to highlight the innovation of this study, and added descriptions of the interpretation of the results, as well as discussions of the results.

**Referee: 2**

*Introduction*

**(1)The first paragraph is too short to introduce the background and I find a weak linkage between the two sentences?**

**Response:**

Thank you very much for your comments. We have made significant modifications to the introduction section, not only adding background information but also highlighting the innovation of this study.

We have added a detailed description of previous studies and pointed out the uncertainties: "The changing correlation between climate and vegetation has recently gained some attention. For example, Wang and Yan (2021) found that the correlation between vegetation and temperature has weakened throughout China over the past 34 years. The precipitation threshold required for vegetation growth in Australia had been found to decrease from 1982 to 2010 (Ukkola et al., 2016). Keenan and Riley (2018) measured how vegetation cover responded to temperature changes and found that the limitations imposed by temperature had decreased over time. Zhao and Yu (2021) found an increased association between climate change and vegetation index variation in Northwest China over the past 34 years. However, most of these studies have solely identified the occurrence of the changes in the relationship between precipitation (or temperature) and vegetation. Uncertainties remain regarding the drivers and how they regulate the changes in the relationship"

The diverse response of vegetation growth to climate variables across land surfaces has indeed always been a hot topic. However, variations may also exist in the patterns of changes in the relationship between climate and vegetation across distinct types of land cover, and the mechanisms behind them are also different, which has been confirmed by our results, but this area of research has not received enough

attention yet. Therefore, we provided a detailed description in the introduction: ”
Vegetation greenness patterns display high spatial heterogeneity across different land surfaces (Gao et al., 2017; Wang et al., 2021), and it's response to climate also varies greatly among different terrestrial ecosystems (Yuan et al., 2019a). The influencing mechanisms of vegetation dynamics in diverse vegetation types have been well documented (Cai et al., 2021; Li et al., 2019; Luo and Chen, 2013; Tao et al., 2015; Upgupta et al., 2015; van Oijen et al., 2018; Wu et al., 2021). Based on previous research, it can be easily inferred that variations may also exist in the patterns of changes in the relationship between climate and vegetation across distinct types of land cover, and the mechanisms behind them are also different, but this area of research has not received enough attention yet”

In addition, we added an explanation in the introduction about why Northwest China was selected as the study region: “Northwest China is characterized by vast areas with different land cover types, including grasslands, forests, and barren lands with sparse vegetation. Since the early 1980s, several studies have indicated warmer and more humid conditions in this area (Liu et al., 2013; Shi et al., 2002; Shi et al., 2007; Wang et al., 2020; Wang et al., 2007; Zhang et al., 2021; Zheng et al., 2021). Recent decades have also seen significant changes in the growth of vegetation in this region (Chen et al., 2019; Niu et al., 2019). As a result, Northwest China presents an ideal opportunity for examining the changes in relationship between climate and vegetation across a variety of vegetation types”

**(2) Line 60 It's difficult to know what is the "relationship between vegetation and climate" represents here. The long-term trend or the interannual variability of climate? for vegetation greenness? Productivity? Growth? Or others? Also, ' As climate varies with climate ', it's not clear "climate" for what**
**Response:**
Thank you very much for your comments. We have made revisions to all these uncertain parts in the introduction.

**(3) The previous efforts on the study topic, the knowledge gap as well as the aim of this study are very ambiguous in Line 60-69. The authors highlight the potential problems in time scales, different time periods and vegetation types, but it's confusing these problems for what? For example, I don't know what are the "different periods", what are the "different vegetation types", and what is the "study period"**
**Response:**
Thank you very much for your comments. We have added a detailed description of previous efforts and removed all of ambiguous statements. We have summarized the deficiencies of previous research: (1). Although the changes in the correlation between climate (precipitation or temperature) and vegetation greenness has recently gained some attention, uncertainties remain regarding the drivers and how they

regulate the changes in the relationship. (2). The diverse response of vegetation growth to climate variables across land surfaces has indeed always been a hot topic. However, ==variations may also exist in the patterns of changes in the relationship between climate and vegetation across distinct types of land cover, and the mechanisms behind them are also different, but this area of research has not received enough attention yet==

**(4)Given that this research topic is not new, they didn't clarify their improvements or their novelty in this study.**

**Response:**

We are sorry for not adequately describing the background and innovation in the introduction earlier. Now we have made significant modifications to the introduction section, not only adding background information but also clarifying our novelty of this study.

*Methods*

**(5)I can't understand why they use the combined NDVI from GIMMS and MODIS datasets because as far as I know, the GIMMS NDVI has released data at least to the end of 2018.**

**(6)It's not clear how to combine the two NDVI datasets. The authors only say that they used the pixel-wise linear regression but how to realize it in detail? They need to provide more details to show how they cope with the trend and variability of the newly combined NDVI time series. From Fig. 2, I can only see they compare the pixels of expanded NDVI with the GIMMS NDVI and MODIS NDVI for the overlapped years but this figure can't verify the time series of the expanded NDVI are credible for their long-term trend and variability.**

**Response:**

Thank you very much for your comments. To avoid errors caused by combining different datasets, we are currently only using the GIMMS NDVI for our research

**(7)As the previous study (Frankenberg et al., 2020, science) points out, there are systematic biases in the AVHRR for the pre-2000 time series, the authors should take caution with the interpretation of potential change before and after 2000 when using GIMMS NDVI**

**Response:**

Thank you very much for your comments. Instead of comparing the NDVI and climate relationship between pre- and post-2000, we now use multiple sliding windows to emphasize the inter-annual variability of this relationship.

**(8)The authors didn't state if the climate and vegetation data were detrended**

**prior to calculating the correlations to avoid the issues like independence of the statistical analysis. If not, I think their results are not credible**

**Response:**

Thank you very much for your comments. We have detrended all variables before studying the vegetation-climate relationship to prevent statistical analysis independence, and the description was stated in Methodology.

*Results*

**(9) The authors completely fail to interpret their results because they didn't provide any data to support what they claimed in the result section. For example, the values of the temporal trend, the values of the correlation, etc**

**Response:**

Thank you very much for your comments. We have added description that interpret our results, for example, this paragraph for the results below: "The statistics categorized by vegetation types show that, $R_{NDVI-GP}$ and $R_{NDVI-GT}$ have shown a declining trend in approximately 81% and 71% of the entire forest area in northwest China, respectively. In the grassland, $R_{NDVI-GP}$ exhibits almost an equal distribution area between positive and negative trends in terms of area, whereas $R_{NDVI-GT}$ displays an upward trend in the majority of the area, accounting for approximately 56%. As for the barren land, the areas occupied by the positive and negative trends of $R_{NDVI-GP}$ or $R_{NDVI-GT}$ are roughly equal."

[Figure]

**Figure 5. Spatial distribution of trends in $R_{NDVI-GP}$ and $R_{NDVI-GT}$ at a 13-year sliding window.** Here, $R_{NDVI-GP}$ is the partial correlation coefficient between GS NDVI and GS precipitation from Interp (**a**), CRU (**c**), and ERA5 (**e**); $R_{NDVI-GT}$ is the partial correlation coefficient between GS NDVI and GS temperature from Interp (**b**), CRU (**d**), and ERA5 (**f**). They are calculated using a 13-year sliding window during 1982-2015. All variables are detrended. The

dots indicates the regions with significant trend in $R_{NDVI-GP}$ (or $R_{NDVI-GT}$) ($P < 0.05$)

**(10)They only show the results from their figures but they never explain why those findings they got, what is mechanisms underlying their findings, and any literature to support their findings. Therefore, I don't know if their findings are robust and reliable. I think they need to discuss their results by comparing with previous studies**

**Response:**

Thank you very much for your comments. We have added sentences for supporting our findings by literature for example, this paragraph for the results below: "The relationship between temperature, precipitation, and NDVI has a close connection with their spatial distribution in the northwest region of China. Generally, when precipitation is low, there is a strong correlation between NDVI and precipitation. However, as precipitation increases, the correlation between NDVI and precipitation gradually weakens, and even becomes negative (Figure 3a, b, c). This is because in areas with water shortages, vegetation growth is highly dependent on precipitation, whereas in relatively humid areas, vegetation growth is less dependent on precipitation (Huxman et al., 2004; Maurer et al., 2020). This also makes the correlation between NDVI and precipitation stronger in areas with sparse vegetation and bare ground compared to grassland and forest. When the temperature is low, the correlation between NDVI and temperature increases as temperature increases. However, when the temperature reaches a certain value, the correlation between NDVI and temperature decreases as the temperature continues to increase (Figure 3d, e, f), indicating that current enhanced greening trend will decline or even disappear (Piao et al., 2006). Previous studies have shown that vegetation growth usually has an optimal temperature, which is the temperature at which vegetation growth is most favorable (Huang et al., 2019; Xu et al., 2013). In areas with low temperatures, vegetation growth is highly dependent on temperature, while in areas with high temperatures, temperature increase can hinder vegetation growth. Therefore, grassland in high-altitude areas is positively correlated with temperature, while sparse vegetation on bare ground in areas with high temperatures is negatively correlated with temperature"

[Figure]

**Figure 3. Scatter plots between R$_{NDVI-GP}$ and precipitation from Interp (a), CRU (b) and ERA5 (c), and scatter plots between R$_{NDVI-GT}$ and temperature from Interp (d), CRU (e) and ERA5 (f).** Here, R$_{NDVI-GP}$ is the partial correlation coefficient between GS NDVI and GS precipitation, and R$_{NDVI-GT}$ is the partial correlation coefficient between GS NDVI and GS temperature (All variables are detrended). The yellow dashed line is the quadratic fit for all the scatters.

In addition, we also conducted a mechanism analysis to investigate how drought regulates the relationship between vegetation and precipitation (temperature) in Discussion. Considering that the decreased temperature sensitivity of tree growth may be attributed to the potential impact of increased drought stress, which can alter the response of plant growth to temperature change (D'Arrigo et al., 2004; Piao et al., 2006) . Reduce in soil water and increase in atmospheric aridity caused by drought will constrain vegetation growth by affecting plant photosynthesis (Piao et al., 2014; Yuan et al., 2019b). Hence, we focus on two mechanisms of vapor pressure deficit (VPD) and soil water volume (SWV), as we examine how their trends regulate the relationship between NDVI and temperature or precipitation. The results are shown below.

[Figure]

**Figure 6. Average trend of R$_{NDVI-GP}$ and R$_{NDVI-GT}$ at a 13-year sliding window in a climate phrase space.** Here, R$_{NDVI-GP}$ is the partial correlation coefficient between GS NDVI and GS precipitation from Interp (**a**), CRU (**b**), and ERA5 (**c**); R$_{NDVI-GT}$ is the partial correlation coefficient between GS NDVI and GS temperature from Interp (**d**), CRU (**e**), and ERA5 (**f**). They are calculated using a 13-year sliding window during 1982-2015. All variables are detrended. The climate space is delineated by changes in GS soil water volume (SWV) and changes in vapor pressure deficit (VPD).

---

## Author Comment (AC3)

**Responses and Changes to Reviewers' Comments**

**Dear Reviewers,**

We really appreciate your helpful suggestions and comments. We have carefully revised the manuscript and addressed all comments. In terms of content, we mainly increased the experimentation of CRU and ERA5 meteorological data to enhance the reliability of the article results. We also conducted a mechanism analysis to investigate how drought regulates the relationship between vegetation and precipitation (temperature). As for the methodology, we detrended all variables before studying the vegetation-climate relationship to prevent statistical analysis independence. Instead of comparing the NDVI and climate relationship between pre- and post-2000, we used multiple sliding windows to emphasize the inter-annual variability of this relationship. In writing, we improved the language use throughout the entire article, made significant changes to the introduction to highlight the innovation of this study, and added descriptions of the interpretation of the results, as well as discussions of the results.

**Referee: 3**

*Major*

**1 The discussion part emphasizes that the human factor is very important, but the draft spent most of the time discussing temperature and precipitation impacts on vegetation. It's better to highlight the key message author wants to convey**

**Response:**

Thank you very much for your suggestions. Considering that the relationship between NDVI and social-economic divers such as GDP and population is very complex that should not be the focus of this study, we removed this part of the content in the article. In the discussion section, we analyzed how drought regulates the relationship between NDVI and precipitation (or temperature) based on two indicators, vapor pressure deficit (VPD) and soil water volume (SWV), and the results with a 13-year sliding window are shown below. We found that the fluctuations in $R_{NDVI-GP}$ and $R_{NDVI-GT}$ coincides closely with the variations in drought conditions. In the areas with the trend in VPD less than 0.02 hpa/yr, where grasslands are predominantly distributed, an increase in SWV tends to cause a decrease in $R_{NDVI-GP}$, but an increase in $R_{NDVI-GT}$. However, when the VPD trend exceeds 0.02 hPa/yr, a more negative trend in SWV tends to result in more negative trends in both $R_{NDVI-GP}$ and $R_{NDVI-GT}$.

[Figure]

**Figure 6. Average trend of RNDVI-GP and RNDVI-GT at a 13-year sliding window in a climate phrase space.** Here, $R_{NDVI-GP}$ is the partial correlation coefficient between GS NDVI and GS precipitation from Interp (**a**), CRU (**b**), and ERA5 (**c**); $R_{NDVI-GT}$ is the partial correlation coefficient between GS NDVI and GS temperature from Interp (**d**), CRU (**e**), and ERA5 (**f**). They are calculated using a 13-year sliding window during 1982-2015. All variables are detrended. The climate space is delineated by changes in GS soil water volume (SWV) and changes in vapor pressure deficit (VPD).

**2 The analysis method is not solid. For example, when mentioning significance, I have no idea how to measure it**

**Response:**

Thank you very much for your suggestions. To avoid disagreements, we removed the normalized linear regression analysis and only studied the relationship between NDVI and climate through the partial correlation analysis. The partial correlation coefficient and significance were described in Methodology and shown below.

The calculation of the partial correlation coefficient is shown in formula (5) (Baba et al., 2004).

$$R_{x_j y} = \frac{-c_{jy}}{\sqrt{c_{jj} c_{yy}}} \tag{5}$$

where $R_{x_j y}$ is the partial correlation coefficient between the $j$th independent variable and the dependent variable $y$, $c$ is the corresponding element in the inverse matrix of the correlation coefficient matrix. The statistical significance of the partial correlations was calculated using the t-test shown in formula(6) (Song and Ma, 2011) with the significance level set to 0.05:

$$t_{xy,z} = \frac{r_{xy,z}}{\sqrt{1 - r_{xy,z}^2}} \sqrt{n - m - 1}. \tag{6}$$

Here, $m$ is the number of independent variables and $n$ is the number of samples.

**3 All results have no quantitative description but only a qualitative description**

**Response:**

Thank you very much for your comments. We have added quantitative description that interpret our results, for example, this paragraph for the results below: "The statistics categorized by vegetation types show that, $R_{NDVI-GP}$ and $R_{NDVI-GT}$ have shown a declining trend in approximately 81% and 71% of the entire forest area in northwest China, respectively. In the grassland, $R_{NDVI-GP}$ exhibits almost an equal distribution area between positive and negative trends in terms of area, whereas $R_{NDVI-GT}$ displays an upward trend in the majority of the area, accounting for approximately 56%. As for the barren land, the areas occupied by the positive and negative trends of $R_{NDVI-GP}$ or $R_{NDVI-GT}$ are roughly equal."

**4 There is no discussion behind the results**

**Response:**

Thank you very much for your comments. We added a discussion behind the results. Firstly, We conducted a mechanism analysis to investigate how drought regulates the relationship between vegetation and precipitation (temperature), and the results are shown below.

[Figure]

**Figure 6. Average trend of $R_{NDVI-GP}$ and $R_{NDVI-GT}$ at a 13-year sliding window in a climate phrase space.** Here, $R_{NDVI-GP}$ is the partial correlation coefficient between GS NDVI and GS precipitation from Interp (**a**), CRU (**b**), and ERA5 (**c**); $R_{NDVI-GT}$ is the partial correlation coefficient between GS NDVI and GS temperature from Interp (**d**), CRU (**e**), and ERA5 (**f**). They are calculated using a 13-year sliding window during 1982-2015. All variables are detrended. The climate space is delineated by changes in GS soil water volume (SWV) and changes in vapor pressure deficit (VPD).

In addition, we performed quadratic curve fitting of $R_{NDVI-GT}$ and time for different vegetation types (shown below), confirming the hypothesis that the impact of rising temperature on the current enhanced greening trend will decline or even disappear

(Piao et al., 2006). We found the forest in northwest China reaches its maximum value of $R_{NDVI-GT}$ earlier than grassland and barren land, suggesting that the decline in the relationship between forest and temperature occurs earlier than it does for grassland and barren land

[Figure]

**Figure 7. Annual variations and quadratic fitting trend lines of $R_{NDVI-GT}$ for different vegetation types.** Here $R_{NDVI-GT}$ is the partial correlation coefficient between GS NDVI and GS temperature. All variables are detrended. Three types of climate data (Interp, CRU and ERA5) are used, corresponding to three columns. The first to fourth lines correspond to the sliding windows of 9years, 11years, 13years and 15years respectively. Dashed lines represent the quadratic curves, while solid lines represent the central lines of quadratic curves.

*Minor*
**The landcover map is not given, it can be added to Figure 1. Also. There may be problems with the NDVI of the bare land**
**Response:**
Thank you very much for your comments. We added a landcover map in Figure1. The areas with an average annual NDVI of less than 0.1 were removed from the analysis (Chen et al., 2018; Piao et al., 2005). Due to the NDVI mask processing, barren land can also be understood as sparsely vegetated areas.

**How to synthesize MODIS (2000-2019) and GIMMS (1982-2015)? It is not stated clearly in the method section**
**Line 80: MODIS NDVI from 2000-2019 and GIMMS NDVI from 1982-2015, but this sentence is not clear**

**What are the advantages and disadvantages of using MODIS and GIMMS in studying the response of vegetation to climate change in these areas? Please provide some references**

**Response:**

Thank you very much for your comments. To avoid errors caused by combining different datasets, we are currently only using the GIMMS NDVI for our research

**Figure 5 and Figure 6 are hard to read, and simplifying them will be better**

**Response:**

Thank you very much for your comments. We removed the content of normalized linear regression analysis in the article

**Fig11 f, y-axis mislabeled**

**Fig12 How to derive GDP in no man's land (e.g., areas in northern Xinjiang)**

**Response:**

Thank you very much for your comments. Considering that the relationship between NDVI and social-economic divers such as GDP and population is very complex that should not be the focus of this study, we removed this part of the content in the article.

**In all figures, how to define no data areas? If you masked NDVI below a certain threshold, please note them in the figure and method**

**Response:**

Thank you very much for your comments. We defined no data areas as the parts with NDVI<0.1. We have noted the no data areas in both figure and method

**Line 85: Please provide the source link; this is not the source from which GIMMS is.**

**Response:**

Thank you very much for your comments. We have provided the source link in the parts of data description

**Line 89: there should be a citation for the MVC method**

Thank you very much for your comments. We added a citation for the MVC method

**Lines 114 and 124: please cite the original author**

Thank you very much for your comments. We have corrected them in the article

**Do not add transparency to the color of the figure4. It is better to set the alpha value to 1, and the leftmost triangle and rectangle of the color bar are separated.**

Thank you very much for your comments. We have corrected them in the article

**Line 194-196: rephrase and it is hard to understand**

**Line 241-243: delete this sentence.**

Thank you very much for your comments. We have rewritten these parts in results

**Line 274: how to calculate significance**

The statistical significance of the partial correlations was calculated using the t-test shown in formula(6) (Song and Ma, 2011) with the significance level set to 0.05:

$$t_{xy,z} = \frac{r_{xy,z}}{\sqrt{1-r_{xy,z}^2}}\sqrt{n-m-1}. \tag{6}$$

Here, $m$ is the number of independent variables and $n$ is the number of samples